# An Optimization-based Approach To Node Role Discovery in Networks: Approximating Equitable Partitions

**Michael Scholkemper**
Department of Computer Science
RWTH Aachen University
scholkemper@cs.rwth-aachen.de

**Michael T. Schaub**
Department of Computer Science
RWTH Aachen University
schaub@cs.rwth-aachen.de

## Abstract

Similar to community detection, partitioning the nodes of a network according to their structural roles aims to identify fundamental building blocks of a network. The found partitions can be used, e.g., to simplify descriptions of the network connectivity, to derive reduced order models for dynamical processes unfolding on processes, or as ingredients for various graph mining tasks. In this work, we offer a fresh perspective on the problem of role extraction and its differences to community detection and present a definition of node roles related to graph-isomorphism tests, the Weisfeiler-Leman algorithm, and equitable partitions. We study two associated optimization problems (cost functions) grounded in ideas from graph isomorphism testing and present theoretical guarantees associated with the solutions to these problems. Finally, we validate our approach via a novel "role-infused partition benchmark", a network model from which we can sample networks in which nodes are endowed with different roles in a stochastic way.

## 1 Introduction

Networks are a powerful abstraction for a range of complex systems [33, 45]. To comprehend such networks we often seek patterns in their connections, e.g., core-periphery structures or densely knit communities. A complementary notion to community structure is that of a *role* partition of the nodes. The concept of node roles, or node equivalences, originates in social network analysis [24]. Node roles are often related to symmetries or connectivity features that can be used to simplify complex networks. Contrary to communities, even nodes that are far apart or are part of different connected components of a network can have the same role [40].

Traditional approaches to defining node roles, put forward in the context of social network analysis [8], consider *exact node equivalences*, based on structural symmetries within the graph structure. The earliest notion is that of *structural equivalence* [26], which assigns the same role to nodes if they are adjacent to the same nodes. Another definition is that of *automorphic equivalence* [14], which states that nodes are equivalent if they belong to the same automorphism orbits. Closely related is the idea of *regular equivalent* nodes [48], defined recursively as nodes that are adjacent to equivalent nodes.

However, large real-world networks often manifest in such a way that these definitions result in a vast number of different roles. What's more, the above definitions do not define a similarity metric between nodes, and it is thus not obvious how to compare two nodes that are deemed *not equivalent*. For example, the above definitions all have in common that nodes with different degrees also have different roles. With the aim of reducing a network's complexity, this is detrimental.

To resolve this problem in a principled way and to provide an effective partitioning of large graphs into nodes with similar roles, we present a quantitative definition of node roles in this paper. Our definition of roles is based on so-called *equitable partitions* (EPs), which are strongly related to the notion of regular equivalence [48]. Crucially, this not only allows us to define an equivalence, but we can also quantify the deviation from an exact equivalence numerically. Further, the notion of EPs generalizes orbit partitions induced by automorphic equivalence classes in a principled way

37th Conference on Neural Information Processing Systems (NeurIPS 2023).

and thus remains tightly coupled to graph symmetries. Knowledge of EPs in a particular graph can, e.g., facilitate the computation of network statistics such as centrality measures [41]. As they are associated with certain spectral signatures, EPs are also relevant for the study of dynamical processes on networks such as cluster synchronization [34, 42], consensus dynamics [51], and network control problems [28]. They have even been shown to effectively imply an upper bound on the expressivity of Graph Neural Networks [30, 50].

**Related Literature** The survey by Rossi and Ahmed [38] puts forward an application-based approach to node role extraction that evaluates the node roles by how well they can be utilized in a downstream machine learning task. However, this perspective is task-specific and more applicable to node embeddings based on roles rather than the actual extraction of roles.

Apart from the already mentioned *exact* node equivalences originating from social network analysis, there exist numerous works on role extraction, which focus on identifying nodes with *similar* roles, by associating each node with a feature vector that is independent of the precise location of the nodes in the graph. These feature vectors can then be clustered to assign nodes to roles. A recent overview article [40] puts forward three categories: First, graphlet-based approaches [35, 39, 25] use the number of graph homomorphisms of small structures to create node embeddings. This retrieves extensive, highly local information such as the number of triangles a node is part of. Second, walk-based approaches [3, 11] embed nodes based on certain statistics of random walks starting at each node. Finally, matrix-factorization-based approaches [17, 20] find a rank-$r$ approximation of a node feature matrix ($F \approx MG$). Then, the left side multiplicand $M \in \mathbb{R}^{|V| \times r}$ of this factorization is used as a soft assignment of the nodes to $r$ clusters.

Jin et al. [21] provide a comparison of many such node embedding techniques in terms of their ability to capture exact node roles such as structural, automorphic, and regular node equivalence. Detailed overviews of (exact) role extraction and its links to related topics such as block modeling are also given in [9, 10].

**Contribution** Our main contributions are as follows:

- We provide a principled stochastic notion of node roles, grounded in equitable partitions, which enables us to rigorously define node roles in complex networks.
- We provide a family of cost functions to assess the quality of a putative role partitioning. Specifically, using a depth parameter $d$, we can control how much of a node's neighborhood is taken into account when assigning roles.
- We present algorithms to minimize the corresponding optimization problems and derive associated theoretical guarantees.
- We develop a generative graph model that can be used to systematically test the recovery of roles in numerical experiments, and use this novel benchmark model to test our algorithms and compare them to well-known role detection algorithms from the literature.

## 2 Notation and Preliminaries

**Graphs.** A simple graph $G = (V, E)$ consists of a node set $V$ and an edge set $E = \{uv \mid u, v \in V\}$. The neighborhood $N(v) = \{x \mid vx \in E\}$ of a node $v$ is the set of all nodes connected to $v$. We allow self-loops $vv \in E$ and positive edge weights $w : E \to \mathbb{R}_+$.

**Matrices.** For a matrix $M$, $M_{i,j}$ is the component in the $i$-th row and $j$-th column. We use $M_{i,\_}$ to denote the $i$-th row vector of $M$ and $M_{\_,j}$ to denote the $j$-th column vector. $\mathbb{I}_n$ is the identity matrix and $\mathbb{1}_n$ the all-ones vector, both of size $n$ respectively. Given a graph $G = (V, E)$, we identify the node set $V$ with $\{1, \dots, n\}$. An *adjacency matrix* of a given graph is a matrix $A$ with entries $A_{u,v} = 0$ if $uv \notin E$ and $A_{u,v} = w(uv)$ otherwise, where we set $w(uv) = 1$ for unweighted graphs for all $uv \in E$. $\rho(A)$ denotes the largest eigenvalue of the matrix $A$.

**Partitions.** A node partition $C = (C_1, C_2, ..., C_k)$ is a division of the node set $V = C_1 \dot\cup C_2 \dot\cup \cdots \dot\cup C_k$ into $k$ disjoint subsets, such that each node is part of exactly one *class* $C_i$. For a node $v \in V$, we write $C(v)$ to denote the class $C_i$ where $v \in C_i$. We say a partition $C'$ is coarser than $C$ ($C' \sqsupseteq C$) if $C'(v) \neq C'(u) \implies C(v) \neq C(u)$. For a partition $C$, there exists a partition indicator matrix $H \in \{0, 1\}^{|V| \times k}$ with $H_{i,j} = 1 \iff i \in C_j$.

## 2.1 Equitable Partitions.

An equitable partition (EP) is a partition $C = (C_1, C_2, ..., C_k)$ such that $v, u \in C_i$ implies that

$$\sum_{x \in N(v)} [C(x) = C_j] = \sum_{x \in N(u)} [C(x) = C_j] \tag{1}$$

for all $1 \leq j \leq k$, where the Iverson bracket $[C(x) = C_j]$ is 1 if $C(x) = C_j$ and 0 otherwise. The coarsest EP (cEP) is the equitable partition with the minimum number of classes $k$. A standard algorithm to compute the cEP is the so-called Weisfeiler-Leman (WL) algorithm [47], which iteratively assigns a color $c(v) \in \mathbb{N}$ to each node $v \in V$ starting from a constant initial coloring. In each iteration, an update of the following form is computed:

$$c^{t+1}(v) = \text{hash}\left(c^t(v), \{\{c^t(x) | x \in N(v)\}\}\right) \tag{2}$$

where hash is an injective hash function, and $\{\{\cdot\}\}$ denotes a multiset (in which elements can appear more than once). In each iteration, the algorithm splits classes that do not conform with eq. (1). At some point $T$, the partition induced by the coloring no longer changes and the algorithm terminates returning the cEP as $\{(c^T)^{-1}(c^T(v)) | v \in V\}$. While simple, the algorithm is a powerful tool and is used as a subroutine in graph isomorphism testing algorithms [4, 29].

On a node level, the cEP carries an immense amount of information about the structure of the network surrounding each node. One can think of it as iteratively building a tree of possible paths from the start node to neighboring nodes. At the first iteration, the color $c^1(v)$ encodes the degree of node $v$ in the graph. At the second iteration, it encodes the number of nodes with a certain degree that are adjacent to $v$. Thus, iterating this update rule (eq. (2)) encodes the structure of possible paths surrounding each node and if these trees are isomorphic for two nodes, the two remain in the same class. This tree of possible paths (also called *unraveling* or *unrolling*) is profoundly important for processes that move over the edges of a network such as dynamical systems [42, 51] and GNNs [50, 30], and even simple ones like PageRank [41] and other centrality measures [44].

The above definition is useful algorithmically but only allows us to distinguish between exactly equivalent vs. non-equivalent nodes. To obtain a meaningful quantitative metric to gauge the quality of a partition, the following equivalent algebraic characterization of an EP will be instrumental: Given a graph $G$ with adjacency matrix $A$ and a partition indicator matrix $H_{\text{cEP}}$ of the cEP, it holds that:

$$AH_{\text{cEP}} = H_{\text{cEP}}(H_{\text{cEP}}^\top H_{\text{cEP}})^{-1} H_{\text{cEP}}^\top A H_{\text{cEP}} =: H_{\text{cEP}} A^\pi. \tag{3}$$

The matrix $AH_{\text{cEP}} \in \mathbb{R}^{n \times k}$ counts in each row (for each node) the number of neighboring nodes within each class ($C_i$ for $i = 1, \ldots, k$), which must be equal to $H_{\text{cEP}} A^\pi$ — a matrix in which each row (node) is assigned one of the $k$ rows of the $k \times k$ matrix $A^\pi$. Thus, from any node $v$ of the same class $C_i$, the sum of edges from $v$ to neighboring nodes of a given class $C_k$ is equal to some fixed number — this is precisely the statement of Equation (1). The matrix $A^\pi$ containing the connectivity statistics between the different classes is the adjacency matrix of the *quotient graph*, which can be viewed as a compressed graph with similar structural properties to the original. In particular, the adjacency matrix of the original graph inherits all eigenvalues from the quotient graph, as can be seen by direct computation. Specifically, let $(\lambda, \nu)$ be an eigenpair of $A^\pi$, then $AH_{\text{cEP}}\nu = H_{\text{cEP}} A^\pi \nu = \lambda H_{\text{cEP}}\nu$.

This makes EPs interesting from a dynamical systems point of view: the dominant (if unique) eigenvector is shared between the graph and the quotient graph. Hence, centrality measures such as Eigenvector Centrality or PageRank are predetermined if one knows the EP and the quotient graph [41]. For similar reasons, the cEP also provides insights into the long-term behavior of other (non)-linear dynamical processes such as cluster synchronization [42] or consensus dynamics [51]. Recently, there has been some study on relaxing the notion of "*exactly*" equitable partitions. One approach is to compare the equivalence classes generated by eq. (2) by computing the edit distance of the trees (the unravellings) that are encoded by these classes implicitly [19]. Another way is to relax the hash function (eq. (2)) to not be injective. This way, "buckets" of coarser equivalence classes are created [7]. Finally, using a less algorithmic perspective, one can define the problem of approximating EP by specifying a tolerance $\epsilon$ of allowed deviation from eq. (1) and consequently asking for the minimum number of clusters that still satisfy this constraint [22]. In this paper, we adopt the opposite approach and instead specify the number of clusters $k$ and then ask for the partition minimizing a cost function (section 4) i.e. the *most equitable* partition with $k$ classes. We want to stress that while

similar, none of these relaxations coincide with our proposed approach. For clarity, we would also like to point out that the notion sometimes called *almost* or *externally* equitable partition [2] in the literature is not the same as the *approximately* equitable partition we propose here. The former is an *exactly* equitable partition (eq. (1)) that simply disregards the number of adjacent neighbors within the same class.

## 2.2 The Stochastic Block Model

The Stochastic Block Model (SBM) [1] is a generative network model which assumes that the node set is partitioned into blocks. The probability of an edge between a node $i$ and a node $j$ is then only dependent on the blocks $B(i)$ and $B(j)$. Given the block labels the expected adjacency matrix of a network sampled from the SBM fulfills:

$$\mathbb{E}[A] = H_B \Omega H_B^\top \tag{4}$$

where $H_B$ is the indicator matrix of the blocks and $\Omega_{B(i),B(j)} = \Pr((i,j) \in E(G))$ is the probability with which an edge between the blocks $B(i)$ and $B(j)$ occurs.

For simplicity, we allow self-loops in the network. The SBM is used especially often in the context of community detection, in the form of the *planted partition model*.

In this restriction of the SBM, there is only an *inside* probability $p$ and an *outside* probability $q$ and $\Omega_{i,j} = p$ if $i = j$ and $\Omega_{i,j} = q$ if $i \neq j$. Usually, one also restricts $p > q$, to obtain a homophilic community structure i.e., nodes of the same class are more likely to connect. However, $p < q$ (heterophilic communities) are also sometimes considered.

## 3 Communities vs. Roles

In this section, we more rigorously define "communities" and "roles" and their difference. To this end, we first consider certain extreme cases and then see how we can relax them stochastically. Throughout the paper, we use the term communities synonymously with what is often referred to as *homophilic communities*, i.e., a community is a set of nodes that is more densely connected within the set than to the outside. In this sense, one may think of a *perfect* community partition into $k$ communities if the network consists of $k$ cliques. In contrast, we base our view of the term "role" on the cEP: If $C$ is the cEP, then $C(v)$ is $v$'s *perfect* role. In this sense, the perfect role partition into $k$ roles is present when the network has an exact EP with $k$ classes. The intuition behind *perfect communities* and *perfect roles* can be seen in appendix A.

In real-world networks, such a perfect manifestation of communities and roles is rare. In fact, even if there was a real network with a perfect community (or role) structure, due to a noisy data collection process this structure would typically not be preserved. Hence, to make these concepts more useful in practice we need to relax them. For communities, the planted partition model relaxes this perfect notion of communities of disconnected cliques to a stochastic setting: The *expected adjacency matrix* exhibits perfect (weighted) cliques — even though each sampled adjacency matrix may not have such a clear structure. To obtain a principled stochastic notion of a node role, we argue that a *planted role model* should, by extension, have an exact cEP in expectation:

**Definition 3.1.** Two nodes $u, v \in V$ have the same *stochastic role* if they are in the same class in the cEP of $\mathbb{E}[A]$.

The above definition is very general. To obtain a practical generative model from which we can sample networks with a planted role structure, we concentrate on the following sufficient condition. We define a probability distribution over adjacency matrices such that for a given a role partition $C$, for $x, y \in C_i$ and classes $C_j$ there exists a permutation $\sigma : C_j \to C_j$ such that $\Pr(A_{x,z} = 1) = \Pr(A_{y,\sigma(z)} = 1) \quad \forall z \in C_j$. That is: *two nodes have the same role if the stochastic generative process that links them to other nodes that have a certain role is the same up to symmetry.* Note that if we restrict $\sigma$ to the identity, we recover the SBM. Therefore, we will consider the SBM as our stochastic generative process in the following.

**RIP model.** In line with our above discussion we propose the *role-infused partition* (RIP) model, to create a well-defined benchmark for role discovery, which allows us to contrast role and community

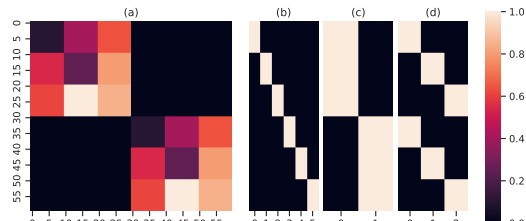

Figure 1: **Example of the RIP model.** It depicts (a) the expected adjacency matrix and the correct according to (b) SBM inference, (c) community detection, (d) role detection.

**Input:** Graph adjacency $A \in \{0,1\}^{n \times n}$,
      number of classes $k$
**Output:** Node assignment $H \in \{0,1\}^{n \times k}$
1   $A = \text{normalize}(A)$
2   Initialize $H = \frac{1}{k}\mathbb{1}_n\mathbb{1}_k^T$
3   **for** *number of steps* **do**
4       $X = AH$
5       $H = \text{cluster}(X)$
6   Return $H$
**Algorithm 1: Approximate Weisfeiler Lehman Algorithm.**

structure. The RIP model is fully described by the parameters $p \in \mathbb{R}, c, k, n \in \mathbb{N}, \Omega_{\text{role}} \in \mathbb{R}^{k \times k}$ as follows: We sample from an SBM with parameters $\Omega, H_B$ (see fig. 1):

$$\Omega_{i,j} = \begin{cases} \Omega_{\text{role}_{i \bmod c, j \bmod c}} & \text{if } \lfloor \frac{i}{c} \rfloor = \lfloor \frac{j}{c} \rfloor \\ p & \text{otherwise} \end{cases} \quad (5)$$

where $H_B$ corresponds to $c \cdot k$ blocks of size $n$. There are $c$ distinct communities - analogous to the planted partition model. The probability of being adjacent for nodes that are not in the same community is $p$. In each community, there are the same $k$ distinct roles with their respective probabilities to attach to one another as defined by $\Omega_{\text{role}}$. Each role has $n$ instances in each community.

Notice that the RIP model has both a planted community structure with $c$ communities and a planted role structure, since $\mathbb{E}[A]$ has an *exact* cEP with $k$ classes (definition 3.1). We stress that the central purpose of our model is to delineate the role recovery from community detection, i.e., community detection is *not* the endeavor of this paper. Rather, the planted communities within the RIP model are meant precisely as an alternative structure that can be found in the data and serve as a control mechanism to determine what structure an algorithm finds. To showcase this, consider Figure 1, which shows an example of the RIP model for $c = 2, k = 3, n = 10, p = 0.1$. It shows a graph that has 2 communities, each of which can be subdivided into the same 3 roles. In standard SBM inference, one would like to obtain 6 blocks - each combination of community and role being assigned its own block (1.b). In community detection, with the objective of obtaining 2 communities, the target clustering would be to merge the first 3 and the second 3 into one cluster respectively (1.c). However, the target clustering for this paper — aiming for 3 roles — is the one on the far right (1.d), combining from each community the nodes that have stochastically the same neighborhood structure.

## 4 Extracting Roles by Approximating the cEP

In this section, we define a family of cost functions (eq. 6, 7) that frame role extraction as an optimization problem. That is, we try to answer the question: *Given a desired number $k$ of role classes, what is the partition that is most similar to an EP?* As discussed above, searching for an *exact* equitable partition with a small number of classes is often not possible: It returns the singleton partition on almost all random graphs [5]. Already small asymmetries, inaccuracies, or noise in data collection can lead to a trivial cEP made up of singleton classes. As such, the cEP is neither a robust nor a particularly useful choice for noisy or even just slightly asymmetric data. Our remedy to the problem is to search for coarser partitions that are closest to being equitable.

Considering the algebraic definition of cEP (eq. 1), intuitively one would like to minimize the difference between the left- and the right-hand side (throughout the paper, we use the $\ell_2$ norm by default and the $\ell_1$ norm where specified):

$$\Gamma_{\text{EP}}(A, H) = ||AH - HD^{-1}H^\top AH|| \quad (6)$$

Here $D = \text{diag}(\mathbb{1}H)$ is the diagonal matrix with the sizes of the classes on its diagonal. We note that $HD^{-1}H^\top = H(H^\top H)^{-1}H^\top = HH^\dagger$ is the projection onto the column space of $H$. However, eq. (6) disregards an interesting aspect of the *exact* cEP. By its definition, the cEP is invariant under multiplication with $A$. That is,

$$A^t H_{\text{cEP}} = H_{\text{cEP}}(A^\pi)^t \quad \text{for all } t \in \mathbb{N}$$

This is especially interesting from a dynamical systems point of view, since dynamics cannot leave the cEP subspace once they are inside it. Indeed, even complex dynamical systems such as Graph Neural Networks suffer from this restriction [50, 30]. To address this, we put forward the following family of cost functions.

$$\Gamma_{d\text{-EP}}(A, H) = \sum_{t=1}^{d} \frac{1}{\rho(A)^t} \Gamma_{\text{EP}}(A^t, H) \tag{7}$$

The factor of $\frac{1}{\rho(A)^i}$ is to rescale the impacts of each matrix power and not disproportionately enhance larger matrix powers. This family of cost functions measures how far the linear dynamical system $t \mapsto A^t H$ diverges from a corresponding equitable dynamical system after $d$ steps. Equivalently, it takes the $d$-hop neighborhood of each node into account when assigning roles. The larger $d$, the *deeper* it looks into the surroundings of a node. Note that all functions of this family have in common that if $H_{\text{cEP}}$ indicates the exact cEP, then $\Gamma_{d\text{-EP}}(A, H_{\text{cEP}}) = 0$ for any choice of $d$.

In the following, we consider the two specific cost functions with extremal values of $d$ for our theoretical results and our experiments: For $d = 1$, $\Gamma_{1\text{-EP}}$ is a measure of the variance of each node's adjacencies from the mean adjacencies in each class (and equivalent to eq. (6)). As such, it only measures the differences in the direct adjacencies and disregards the longer-range connections. We call this the *short-term* cost function. The other extreme we consider is $\Gamma_{\infty\text{-EP}}$, where $d \to \infty$. This function takes into account the position of a node within the whole graph. It takes into account the long-range connectivity patterns around each node. We call this function the *long-term* cost.

In the following sections, we aim to optimize these objective functions to obtain a clustering of the nodes according to their roles. However, when optimizing this family of functions, in general, there exist examples where the optimal assignment is not isomorphism equivariant (see appendix B). As isomorphic nodes have *exactly* the same *global* neighborhood structure, arguably, they should be assigned the same role. To remedy this, we restrict ourselves to partitions compatible with the cEP when searching for the minimizer of these cost functions.

## 4.1 Optimizing the Long-term Cost Function

In this section, we consider optimizing the long-term objective eq. (7). This is closely intertwined with the dominant eigenvector of $A$, as the following theorem shows:

**Theorem 4.1:** *Let $\mathcal{H}$ be the set of indicator matrices $H \in \{0, 1\}^{n \times k}$ s.t. $H \mathbb{1}_k = \mathbb{1}_n$. Let $A \in \mathbb{R}^{n \times n}$ be an adjacency matrix. Assume the dominant eigenvector to the eigenvalue $\rho(A)$ of $A$ is unique. Using the $\ell_1$ norm in eq. (6), the optimizer*

$$OPT = \arg \min_{H \in \mathcal{H}} \lim_{d \to \infty} \Gamma_{d\text{-EP}}(A, H)$$

*can be computed in $\mathcal{O}(a + nk + n \log(n))$, where $a$ is the time needed to compute the dominant eigenvector of $A$.*

The proof of the theorem directly yields a simple algorithm that efficiently computes the optimal assignment for the long-term cost function. Simply compute the dominant eigenvector $v$ and then cluster it using 1-dimensional $k$-means. We call this *EV-based clustering*.

## 4.2 Optimizing the Short-term Cost Function

In contrast to the previous section, the short-term cost function is more challenging. In fact,

**Theorem 4.2:** *Optimizing the short-term cost is NP-hard.*

In this section, we thus look into optimizing the short-term cost function by recovering the stochastic roles in the RIP model. Given $s$ samples $A^{(s)}$ of the same RIP model, asymptotically, the sample mean $\frac{1}{s} \sum_{i=1}^{s} A^{(i)} \to \mathbb{E}[A]$ converges to the expectation as $s \to \infty$. Thus, recovering the ground truth partition is consistent with minimizing the short-term cost in expectation.

To extract the stochastic roles, we consider an approach similar to the WL algorithm which computes the exact cEP. We call this the Approximate WL algorithm (Algorithm 1). A variant of this without the clustering step was proposed in [23]. Starting with one class encompassing all nodes, the algorithm

iteratively computes an embedding vector $x = (x_1, ..., x_k)$ for each node $v \in V$ according to the adjacencies of the classes:

$$x_i = \sum_{u \in N(v)} [C^{(t)}(u) = C_i^{(t)}]$$

The produced embeddings are then clustered to obtain the partition into $k$ classes $H$ of the next iteration. The clustering routine can be chosen freely. This is the big difference to the WL algorithm, which computes the number of classes on-the-fly — without an upper bound to the number of classes. Another difference is that the assignment from the previous iteration is not taken into account. This is a design choice and clustering $X$ concatenated with $H$ works similarly. In practice, we have found that omitting $H$ makes it easier to reassign nodes that were assigned to the wrong cluster in a previous iteration, thus making the algorithm more robust. It also does not take away from the power of the algorithm as Morris et al. [30] show this variant to be equivalent to the original WL algorithm. The main theoretical result of this section uses average linkage for clustering:

**Theorem 4.3:** *Let A be sampled from the RIP model with parameters $p \in \mathbb{R}, c \in \mathbb{N}, 3 \leq k \in \mathbb{N}, n \in \mathbb{N}, \Omega_{role} \in \mathbb{R}^{k \times k}$. Let $H_{role}^{(0)}, ..., H_{role}^{(T')}$ be the indicator matrices of each iteration when performing the exact WL algorithm on $\mathbb{E}[A]$. Let $\delta = \min_{0 \leq t' \leq T'} \min_{i \neq j} ||(\Omega H_{role}^{(t')})_{i,\_} - (\Omega H_{role}^{(t')})_{j,\_}||$. Using average linkage in algorithm 1 in the clustering step and assuming the former correctly infers k, if*

$$n > -\frac{9\mathcal{W}_{-1}((q-1)\delta^2/9k^2)}{2\delta^2} \tag{8}$$

*where $\mathcal{W}$ is the Lambert W function, then with probability at least q: Algorithm 1 finds the correct role assignment using average linkage for clustering.*

The proof hinges on the fact that the number of links from each node to the nodes of any class concentrates around the expectation. Given a sufficient concentration, the correct partitioning can then be determined by the clustering step. Notice that even though we allow for the SBM to have more blocks than there are roles, the number of roles (and the number of nodes therein) is the delimiting factor here — not the overall number of nodes. Notice also that theorem 4.3 refers to *exactly* recovering the partition from only *one* sample. Typically, concentration results refer to a concentration of multiple samples from the same model. Such a result can be derived as a direct consequence of Theorem 4.3 and can be found in appendix E. The bound given in the theorem is somewhat crude in the sense that it scales very poorly as $\delta$ decreases. This is to be expected as the theorem claims exact recovery for all nodes with high probability.

**Fractional Assignments** In a regime, where the conditions of Theorem 4.3 do not hold, it may be beneficial to relax the problem. A hard assignment in intermediate iterations, while possible, has shown to be empirically unstable (see experiments). Wrongly assigned nodes heavily impact the next iteration of the algorithm. As a remedy, a soft assignment — while not entirely different — has proven more robust. We remain concerned with finding the minimizer $H$ of eq. (6) However, we no longer constrain $H_{i,j} \in \{0, 1\}$, but relax this to $0 \leq H_{i,j} \leq 1$. $H$ must still be row-stochastic - i.e. $H\mathbb{1} = \mathbb{1}$. That is, a node may now be *fractionally* assigned to multiple classes designating how strongly it belongs to each class. This remedies the above problems, as algorithms such as fuzzy $c$-means or Bayesian Gaussian Mixture Models are able to infer the number of clusters at runtime and must also not make a hard choice about which cluster a node belongs to. This also allows for Gradient Descent approaches like GNNs. We investigate these thoughts empirically in the experiments section.

## 5   Numerical Experiments

For the following experiments, we use two variants of the Approximate WL algorithm (1), one where the clustering is done using average linkage and one where fuzzy $c$-means is used. We benchmark the EV-based clustering (4.1) and the 2 variants of the Approximate WL algorithm as well as node classes obtained from the role2vec [3] and the node2vec [16], struc2vec [36] and rolX [17] node embeddings (called *R2V, N2V, S2V* and *rolX* in the figures). We retrieve an assignment from the baseline benchmark embeddings by $k$-means. The three X2vec node embedding techniques use autoencoders with skip-gram to compress information obtained by random walks. rolX uses *recursive feature extraction* [18] paired with non-negative matrix factorization. While node2vec is a somewhat

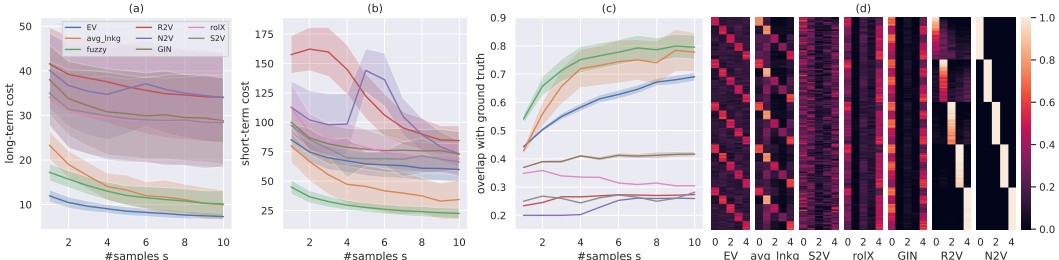

Figure 2: **Role recovery on graphs sampled from the RIP model.** (a-c) On the x-axis, we vary the number of samples $s$ that are averaged to obtain the input $A$. The graphs used are randomly sampled from the planted partition model. On the y-axis, we report the long-term cost $c_{20-\mathrm{EP}}$ (a), the short-term cost (b), and the overlap of the clusterings with the ground-truth (c) over 100 runs along with their standard deviation. In (d), the average role assignment (rows reordered to maximize overlap) is shown for the number of samples $s = 1$.

universal node embedding technique also taking into account the community structure of a network, role2vec, struc2vec, and rolX are focussed on embedding a node due to its role. All embeddings are state-of-the-art node embedding techniques used for many downstream tasks. Further, we compare the above algorithms to the GIN [50], which is trained to minimize the short-term cost individually on each graph. The GIN uses features of size 32 that are uniformly initialized and is trained for 1000 epochs. To enable a fair comparison, we convert the fractional assignments into hard assignments by taking the class with the highest probability for each node. Experimental data and code is available here[1].

**Experiment 1: Planted Role Recovery.**    For this experiment, we sampled adjacency matrices $A^{(i)}$ from the RIP model as described in Section 3 with $c = k = 5, n = 10, p = 0.05, \Omega_{\mathrm{role}} \in \mathbb{R}^{k \times k}$. Each component of $\Omega_{\mathrm{role}}$ is sampled uniformly at random i.i.d from the interval $[0, 1]$. We then sample $s$ samples from this RIP model and perform the algorithms on the sample mean. The mean and standard deviation of long-term and short-term costs and the mean recovery accuracy of the ground truth and its variance are reported in Figure 2 over 100 trials for each value of $s$. The overlap score of the assignment $C$ with the ground truth role assignment $C^{\mathrm{gt}}$ is computed as:

$$\mathrm{overlap}(C, C^{\mathrm{gt}}) = \max_{\sigma \in \mathrm{permutations}(\{1,...,k\})} \sum_{i=1}^{k} \frac{|C_{\sigma(i)} \cap C_i^{\mathrm{gt}}|}{|C_i|}$$

Figure 2 (d) shows the mean role assignments output by each algorithm. Since the columns of the output indicator matrix $H$ of the algorithms are not ordered in any specific way, we use the maximizing permutation $\sigma$ to align the columns before computing the average.

*Discussion.* In Figure 2 (a), one can clearly see that the EV-based clustering outperforms all other algorithms measured by long-term cost, validating Theorem 4.1. While both Approximate WL algorithms perform similarly in the cost function, the fuzzy variant has a slight edge in recovery accuracy. We can see that the tendencies for the short-term cost and the accuracy are directly adverse. The Approximate WL algorithms have the lowest cost and also the highest accuracy in recovery. The trend continues until all three X2vec algorithms are similarly bad in both measures. The GIN performs better than the X2vec algorithms both in terms of cost and accuracy. However, it mainly finds 2 clusters. This may be because of the (close to) uniform degree distribution in these graphs. On the contrary, the node2vec and role2vec detect the communities instead of the roles. This is surprising for role2vec since it aims to detect roles.

**Experiment 2: Inferring the Number of Roles and Centrality.**    A prominent problem in practice that has been scarcely addressed in this paper so far is that the number of roles may not be known. Some algorithms — like fuzzy $c$-means or GIN — can infer the number of clusters while performing the clustering. In this experiment, we consider the *protein* dataset [6] and run the suite of algorithms

---

[1]https://git.rwth-aachen.de/netsci/publication-2023-an-optimization-based-approach-to-node-role-discovery-in-networks

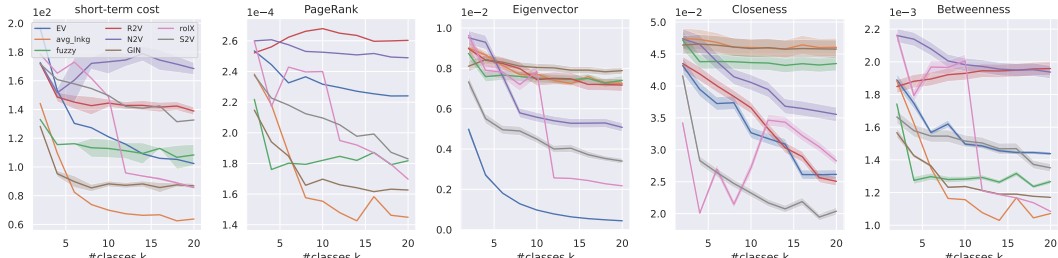

Figure 3: **Recovery of centralities on a real-world network.** On the x-axis, the number of classes $2 \leq k \leq 20$ that the algorithms are tasked to find is shown. On the y-axis, the mean short-term cost $c_{\mathrm{EP}}$, the average deviation from the cluster mean is then shown from left to right for PageRank, Eigenvector centrality, Closeness, and Betweenness over 10 trials on the protein dataset.

for varying $2 \leq k \leq 20$. The mean short-term cost of the assignments and its standard deviation is reported in Figure 3. Additionally for the PageRank, Eigenvector, Closeness, and Betweeness Centrality, the $l_1$ deviations of each node from the mean cluster value are reported.

*Discussion.* In Figure 3, all algorithms show a similar trend. The costs decrease as the number of clusters increases. The elbow method yields $k = 4, ..., 12$ depending on the algorithm. The GIN performs much better than in the previous experiment. This may be due to the fact that there are some high-degree nodes in the dataset, that are easy to classify correctly. The converse is true for the fuzzy variant of Approximate WL that implicitly assumes that all clusters should have about the same size. The EV algorithm clusters the nodes well in terms of Eigenvector centrality, which is to be expected. Both struc2vec and rolX achieve the best results for Closeness, which not captured well by the Approximate WL algorithms. The sudden dip for rolX at $k = 12$ is due to the fact that rolX cannot be applied when the number of clusters is larger than the number of features. We use $k$-means to extract the role assignment in this case.

**Experiment 3: Approximate EP as a Node Embedding.** In this section, we diverge a little from the optimization-based perspective of the paper up to this point and showcase the effectiveness of the information content of the extracted roles in a down-stream learning task. This links our approach to the application-based role evaluation approach of [38] and provides a better comparison with state-of-the-art node embedding techniques. In this experiment, we use an embedding based on our role assignments: For a given adjacency matrix $A$ with dominant eigenvector $\nu$, we compute the output of the Approximate WL algorithm $H$ using average linkage. We also compute the output $H^{\mathrm{EV}}$ of the EV-based clustering (section 4.1). Then the embedding is defined as follows:

$$\mathrm{f}(i) = (H^{\mathrm{EV}}_{i,\_}, (AH^{\mathrm{EV}})_{i,\_}, H_{i,\_}, (AH)_{i,\_}, \nu_i)$$

That is, the embedding is made up of the short- and long-term role assignments and role adjacencies as well as the component of the dominant eigenvector (used for the long-term optimization). We recreate the experimental design of Ribeiro et al. [36, Section 4.4]. The networks are flight networks where the nodes are airports and the edges are flights between them. In each graph, the airports are labeled with a class reflecting their activity, which is the target for this task. We then fit a "one-vs-all" logistic regression and perform a 5-fold cross-validation, tuning hyperparameters for each fold. The results are reported in Table 1.

*Discussion.* Table 1 shows that our embedding and the struc2vec embedding perform best by a margin over the other algorithms. Our embedding has a slight edge in each of the datasets over struc2vec. This shows that the extracted roles provide a good characterization of the roles of the nodes in the networks.

**Experiment 4: Graph Embedding.** We continue the line of thought of the previous experiment and consider graph-level classification tasks instead. Specifically, we consider a setting with only few learning examples. In this setting, the dimensionality of the embedding is extremely important. Therefore, we analyze which part of the above embedding is most important for the classification task, which leads to the following aggregated embedding:

$$\mathrm{F}_{\mathrm{aEP}}(G) = \left(|C_1|, ..., |C_k|, \frac{1}{|C_1|} \sum_{i \in C_1} v_i, ..., \frac{1}{|C_k|} \sum_{i \in C_k} v_i\right)$$

Table 1: **Performance as a node embedding on a classification task.** Mean accuracy in % and standard deviation over a 5-fold cross-validation on the airport networks of [36].

|        | ours | struc2vec | rolX | role2vec | node2vec | deg |
|--------|------|-----------|------|----------|----------|-----|
| brazil | **90.0** $\pm$ 8.5 | 86.4 $\pm$ 3.3 | 78.0 $\pm$ 1.2 | 70.2 $\pm$ 11.1 | 62.8 $\pm$ 14.6 | 69.2 $\pm$ 0.6 |
| usa | **73.8** $\pm$ 0.3 | 72.6 $\pm$ 1.8 | 62.9 $\pm$ 0.5 | 64.6 $\pm$ 1.5 | 59.9 $\pm$ 1.1 | 53.7 $\pm$ 0.0 |
| europe | **79.8** $\pm$ 6.5 | 76.4 $\pm$ 1.5 | 60.8 $\pm$ 0.9 | 48.9 $\pm$ 2.8 | 64.3 $\pm$ 3.9 | 53.6 $\pm$ 0.0 |

Table 2: **Few shot graph embedding performance.** Mean accuracy in % over 10 runs of the EV embedding, Graph2Vec and the GIN. For each run, we randomly sample 10 data points for training and evaluate with the rest. As a comparison, the GIN+ is trained on 90% of the data points.

|          | ours | G2Vec | GIN | GIN+ |
|----------|------|-------|-----|------|
| AIDS     | **95.0** $\pm$ 5.2 | 79.9 $\pm$ 4.4 | 80.0 $\pm$ 14.1 | 97.8 $\pm$ 1.4 |
| ENZYMES  | **21.3** $\pm$ 1.7 | 20.3 $\pm$ 1.5 | 21.0 $\pm$ 1.7 | 60.3 $\pm$ 0.7 |
| PROTEINS | **66.5** $\pm$ 6.4 | 60.3 $\pm$ 3.5 | 59.6 $\pm$ 7.6 | 75.4 $\pm$ 1.3 |
| NCI1     | **58.5** $\pm$ 4.0 | 53.9 $\pm$ 1.5 | 50.0 $\pm$ 1.4 | 82.0 $\pm$ 0.3 |
| MUTAG    | **81.5** $\pm$ 7.7 | 66.9 $\pm$ 5.6 | 77.5 $\pm$ 11.1 | 94.3 $\pm$ 0.5 |

The value for $k$ was found by a grid search over $k \in \{2, ..., 40\}$. We benchmark this against the commonly used graph embedding Graph2Vec [32] and the GIN. We use graph classification tasks from the field of bioinformatics ranging from 188 graphs with an average of 18 nodes to 4110 graphs with an average of 30 nodes. The datasets are taken from [31] and were first used (in order of Table 2) in [37, 43, 13, 46, 12]. In each task, we use only 10 data points to train a 2-layer MLP on the embeddings and a 4-layer GIN. Each hidden MLP and GIN layer has 100 nodes. The Graph2Vec embedding is set to size 16, whereas the GIN receives as embedding the attributes of the nodes of the respective task. The GIN is thus *informed*. We also report the accuracy of a GIN that is trained on 90% of the respective data sets. The results over 10 independent runs are reported in Table 2.

*Discussion.* Experiment 4 has the EV embedding as the overall winner of few-shot algorithms. Our claim here is not that the EV embedding is a particularly powerful few-shot learning approach, but that the embedding carries a lot of structural information. Not only that but it is robust in the sense that few instances are sufficient to train a formidable classifier. However, it pales in comparison with the "fully trained" GIN, which performs better on every dataset.

## 6   Conclusion

This paper has brought forward a rigorous definition of "role" in networks that mirrors the notion of community established before. Based on this definition, we presented the RIP model as a generative model to sample networks with predetermined roles. We also defined quality measures for role assignments and proposed an optimization-based framework for role extraction. Each of the two proposed cost functions measures how well a certain characteristic of the cEP is upheld - long-term or short-term behavior. We proposed an algorithm for finding the optimal clustering for the long-term cost function and related the optimization of the other cost function to the retrieval of stochastic roles from the RIP model.

**Limitations**   The proposed cost functions are sensitive to the degree of the nodes. In scale-free networks, for example, it can happen that few extremely high-degree nodes are put into singleton clusters and the many remaining low-degree nodes are placed into one large cluster. The issue is somewhat reminiscent of choosing a minimal min-cut when performing community detection, which may result in a single node being cut off from the main graph. A remedy akin to using a normalized cut may thus be a helpful extension to our optimization-based approach. Future work may thus consider correcting for the degree, and a further strengthening of our theoretic results.

## Acknowledgements

This research has been supported by the DFG RTG 2236 "UnRAVeL" – Uncertainty and Randomness in Algorithms, Verification and Logic - and the Ministry of Culture and Science of North Rhine-Westphalia (NRW Rückkehrprogramm).

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

# Appendix

## A    Example of Communities vs. Roles

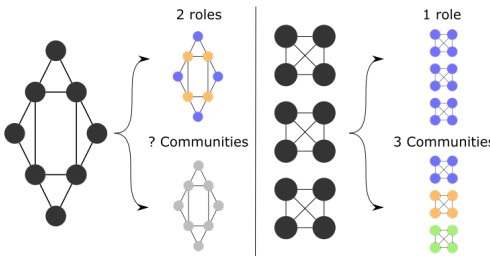

Figure A1: **Toy example** showing two networks and their role/community structure. The left has two *exact* roles but how many communities it has is not clear. The right has 3 *perfect* communities, but only a single role.

## B    Example of Isomorphic Nodes that receive a different role

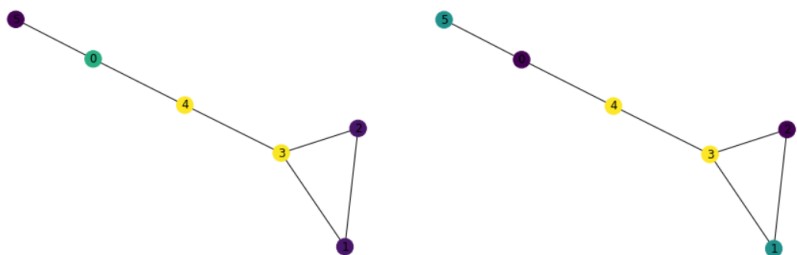

Figure A2: **Example graphs** showing undesirable properties of the optimizer of eq. (6). Nodes 1 and 2 are in the same cEP class; they are even isomorphic. However, the minimizer of eq. (6) puts them into different classes. The partition minimizing eq. (6) as given by the right yields a score of $0.707$, whereas the best partition that respects the cEP yields $0.816$.

## C    Proof of theorem 4.1

**Theorem 4.1:**    *Let $\mathcal{H}$ be the set of indicator matrices $H \in \{0,1\}^{n \times k}$ s.t. $H\mathbb{1}_k = \mathbb{1}_n$. Let $A \in \mathbb{R}^{n \times n}$ be an adjacency matrix. Assume the dominant eigenvector to the eigenvalue $\rho(A)$ of $A$ is unique. Using the $\ell_1$ norm in eq. (6), the optimizer*

$$OPT = \arg\min_{H \in \mathcal{H}} \lim_{d \to \infty} \Gamma_{d\text{-}EP}(A, H)$$

*can be computed in $\mathcal{O}(a + nk + n\log(n))$, where $a$ is the time needed to compute the dominant eigenvector of $A$.*

*Proof.*  Consider the long-term cost function (eq. 6,7):

$$c_{\text{d-EP}}(A, H) = \sum_t^d \frac{1}{\rho(A)^t} ||A^t H - H D^{-1} H^\top A^t H||$$

$$= \sum_t^d ||(\mathbb{I}_n - H D^{-1} H^\top) \frac{1}{\rho(A)^t} A^t H||$$

$$\overset{\lim d \to \infty}{=} ||(\mathbb{I}_n - H D^{-1} H^\top) w v^\top H||$$

We arrive at a formulation akin to the $k$-means cost function. However, $wv^\top H$ is in general not independent of the clustering, as would be the case in the usual formulation of $k$-means. This can be used advantageously by rewriting the above matrix equation element-wise:

$$= \sum_i^n \sum_j^k |w_i(v^\top H_{\_,j}) - \frac{1}{|C(w_i)|} \sum_{l \in C(w_i)} w_l(v^\top H_{\_,j})|$$

$$= \sum_i^n \sum_j^k |(w_i - \frac{1}{|C(w_i)|} \sum_{l \in C(w_i)} w_l)(v^\top H_{\_,j})|$$

It is possible to completely draw out the constant factor of $\sum_j v^\top H_{\_,j} = \sum_i v_i$ since the row sums of $H$ are 1 and the components $v_i \geq 0$ are non-negative.

$$= \sum_i^n \sum_j^k (v^\top H_{\_,j})|(w_i - \frac{1}{|C(w_i)|} \sum_{l \in C(w_i)} w_l)|$$

$$= \text{const} \sum_i^n |(w_i - \frac{1}{|C(w_i)|} \sum_{l \in C(w_i)} w_l)|$$

We end up at a formulation equivalent to clustering $w$ into $k$ clusters using $k$-means. We can now notice that $w$ is only 1-dimensional and as such the $k$-means objective can be optimized in $\mathcal{O}(n \log(n) + nk)$ [49, 15]. $\qquad \square$

# D   Proof of theorem 4.2

**Theorem 4.2:**   *Optimizing the short-term cost is NP-hard.*

*Proof.* We reduce from the PLANAR-K-MEANS problem, which is shown to be NP-hard in [27]. In PLANAR-K-MEANS, we are given a set $\{(x_1, y_1), ..., (x_n, y_n)\}$ of $n$ points in the plane and a number $k$ and a cost $c$. The problem is to find a partition of the points into $k$ clusters such that the cost of the partition is at most $c$, where the cost of a partition is the sum of the squared distances of each point to the center of its cluster. We now formulate the decision variant of optimizing the short-term cost which we show is NP-hard.

**Definition D.1** (K-AEP). Let $G = (V, E)$ be a graph, $k \in \mathbb{N}$ and $c \in \mathbb{R}$. K-AEP is then the problem of deciding whether there exists a partition of the nodes in $V$ into $k$ clusters such that the short-term cost $\Gamma_{1-\text{EP}}$ (eq. (7)) using the squared L2 norm is at most $c$.

Let $W(X, Y)$ be the sum of the weights of all edges between $X, Y \subseteq V$. Additionally, for a given partition indicator matrix $H$, let $C_i$ be the set of nodes $v$ s.t. $H_{v,i} = 1$. For the following proof, the equivalent definition of the short-term cost function (eq. 6) using the squared L2 norm is more convenient:

$$\Gamma_{\text{EP}}(A, H) = \sum_i \sum_j \sum_{v \in C_i} (W(\{v\}, C_j) - \frac{1}{|C_i|} W(C_i, C_j))^2$$

We now show that K-AEP is NP-hard by reduction from PLANAR-K-MEANS.

*Construction.* Given $\{(x_1, y_1), ..., (x_n, y_n)\}$ of $n$ points in the plane and a number $k'$ and a cost $c'$ construct the following graph: We shift the given points by $-\min_{i \in [n]} x_i$ in their $x$-coordinate and by $-\min_{i \in [n]} y_i$ in their $y$-coordinate. This makes them non-negative, but does not change the problem. Let $D = 1 + \sum_{i=1}^n x_i^2 + y_i^2$. Notice that $D$ is an upper bound on the cost of the $k$-means partition. To start, let $V = \{a, b\}$. Add self-loops of weight $3D$ to $a$ and of weight $6D$ to $b$. For each point $(x_i, y_i)$, add a node $m_i$ to $V$ and add edges $m_i a$ of weight $x_i$ and $m_i b$ of weight $y_i$ to $E$.

$G = (V, E), k = k' + 2, c = c'$ are now the inputs to K-AEP.

*Correctness.* We now prove that the PLANAR-K-MEANS instance has a solution if and only if K-AEP has a solution. Assume that PLANAR-K-MEANS has a solution $S' = (S'_1, ..., S'_{k'})$ that has cost $c^* \leq c'$. Then the solution we construct for K-AEP is $S = (S_1, ..., S_{k'}, \{a\}, \{b\})$, where $m_i \in S_j \iff (x_i, y_i) \in S'_j$. The cost of this solution is:

$$c^+ = \sum_{i=1}^{k} \sum_{j=1}^{k} \sum_{v \in S_i} (W(\{v\}, S_j) - \frac{1}{|S_i|} W(S_i, S_j))^2$$

Since $a$ and $b$ are in singleton clusters, their outgoing edges do not differ from the cluster average and so incur no cost. The remaining edges either go from $V \setminus \{a, b\}$ to $a$ or from $V \setminus \{a, b\}$ to $b$. So, the sum reduces to:

$$c^+ = \sum_i \sum_{v \in S_i} \left( (w(v, a) - \frac{1}{|S_i|} W(S_i, \{a\}))^2 + (w(v, b) - \frac{1}{|S_i|} W(S_i, \{b\}))^2 \right)$$

Since $\mu_x(S_i) := \frac{1}{|S_i|} W(S_i, \{a\})$ is the average weight of the edges from $S_i$ to $a$, and these edges have weight according to the $x$ coordinate of the point they were constructed from, $\mu_x(S_i)$ is equal to the mean $x$ coordinate within the cluster $S'_i$. This concludes the proof of this direction, as:

$$c^+ = \sum_{S'_i \in S'} \sum_{(x_l, y_l) \in S'_i} (x_l - \mu_x(S'_i))^2 + (y_l - \mu_y(S'_i))^2 = c^* \leq c'$$

For the other direction, assume we are given a solution $S = (S_1, ..., S_{k+2})$ to K-AEP with cost $c^+ \leq c$. We distinguish two cases:

*Case 1:* $\exists i \in \mathbb{N}$ s.t. $S_i \supsetneq \{a\}$ *or* $S_i \supsetneq \{b\}$. Assume that $S_i \supsetneq \{a\}$, if also $b \in S_i$ then the cost is at least the difference of the two self-loops:

$$c^+ \geq \sum_{v \in S_i} \left( W(\{v\}, S_i) - \frac{1}{|S_i|} W(S_i, S_i) \right)^2$$
$$\geq \left( \frac{1}{2} \max_{u,v \in S_i} W(\{v\}, S_i) - W(\{u\}, S_i) \right)^2$$
$$\geq \left( \frac{1}{2} (w(b, b) - W(\{a\}, S_i)) \right)^2$$
$$\geq \left( \frac{1}{2} (6D - 4D) \right)^2 = D^2 \geq D$$

If instead, some $m_j \in S_i$, then the cost is at least the difference of the self-loop to $a$ and the edge from $m_j$ to $a$:

$$c^+ \geq \sum_{v \in S_i} \left( W(\{v\}, S_i) - \frac{1}{|S_i|} W(S_i, S_i) \right)^2$$
$$\geq \left( \frac{1}{2} (w(a, a) - W(\{m_j\}, S_i)) \right)^2$$
$$\geq \left( \frac{1}{2} (3D - D) \right)^2 = D^2 \geq D$$

Thus $c^+ \geq D$ is so large that any clustering of the points has at most cost $c \geq D$ thus a solution to the PLANAR-K-MEANS instance exists. The case where $S_i \supsetneq \{b\}$ is analogous.

*Case 2: Case 1 doesn't hold.* In this case, we have $S = (S_1, ..., S_k, \{a\}, \{b\})$ which yields a clustering $S' = (S'_1, ..., S'_k)$ for the PLANAR-K-MEANS, where $m_i \in S_j \iff (x_i, y_i) \in S'_j$. This instance has cost $c^+ = c^* \leq c$. $\qquad\square$

# E Proof of theorem 4.3

**Theorem 4.3:** *Let $A$ be sampled from the RIP model with parameters $p \in \mathbb{R}, c \in \mathbb{N}, 3 \leq k \in \mathbb{N}, n \in \mathbb{N}, \Omega_{role} \in \mathbb{R}^{k \times k}$. Let $H_{role}^{(0)}, ..., H_{role}^{(T')}$ be the indicator matrices of each iteration when performing the exact WL algorithm on $\mathbb{E}[A]$. Let $\delta = \min_{0 \leq t' \leq T'} \min_{i \neq j} ||(\Omega H_{role}^{(t')})_{i,\_} - (\Omega H_{role}^{(t')})_{j,\_}||$. Using average linkage in algorithm 1 in the clustering step and assuming the former correctly infers $k$, if*

$$n > -\frac{9\mathcal{W}_{-1}((q-1)\delta^2/9k^2)}{2\delta^2} \tag{8}$$

*where $\mathcal{W}$ is the Lambert W function, then with probability at least q: Algorithm 1 finds the correct role assignment using average linkage for clustering.*

*Proof.* Consider the adjacency matrix $A_B$ of a simple binomial random graph of size $n$ - i.e. a single block of the SBM. Let $\delta^* < \frac{\delta}{3}$. Using the Chernoff bound for binomial random variables, we have that the degree of a single node $i$ is within the ball of size $\delta^*$ with probability:

$$\Pr\left(|(A_B \mathbb{1})_i - (\mathbb{E}[(A_B \mathbb{1})_i]| \geq \delta^* \cdot n\right) \leq 2e^{-2n(\delta^*)^2}$$

The probability that all nodes fall in close proximity to the expectation, is then simply:

$$\Pr\left(\|A_B \mathbb{1} - \mathbb{E}[A_B \mathbb{1}]\|_\infty \geq \delta^* \cdot n\right) \leq \left(1 - 2e^{-2n(\delta^*)^2}\right)^n$$

Finally, in the SBM setting, we have $k^2$ such blocks and the probability that none of the nodes are far away from the expectation in any of these blocks is:

$$\Pr\left(\frac{\left\|AHH_{role}^{(T)} - \mathbb{E}[AHH_{role}^{(T)}]\right\|}{n} \geq \delta^*\right) \leq \left(1 - 2e^{-2n(\delta^*)^2}\right)^{nk^2}$$

We can upper bound this by its first-order Taylor approximation:

$$\left(1 - 2e^{-2n(\delta^*)^2}\right)^{nk^2} \leq p \leq 1 - 2nk^2 e^{-2n(\delta^*)^2}$$

$$\Leftrightarrow \qquad \frac{(p-1)(\delta^*)^2}{k^2} \leq -2n(\delta^*)^2 e^{-2n(\delta^*)^2}$$

$$\Leftrightarrow \qquad \mathcal{W}_{-1}\left(\frac{(p-1)(\delta^*)^2}{k^2}\right) \geq -2n(\delta^*)^2$$

$$\Leftrightarrow \qquad -\frac{9\mathcal{W}_{-1}((p-1)\delta^2/9k^2)}{2\delta^2} \leq n$$

Thus with probability at least $p$, the maximum deviation from the expected mean is $\delta^*$, which is why we simply assume this to be the case going forward, i.e.:

$$\frac{1}{n} \max_{i,j} \left(\left|\left(AHH_{role}^{(T)} - \mathbb{E}[AHH_{role}^{(T)}]\right)_{i,j}\right|\right) < \frac{\delta}{3}$$

Consider the L1 distance of nodes inside the same cluster: This is at most $k\frac{\delta}{3}$. For nodes that belong to different clusters, this will be at least $k(\delta - 2\delta^*) > k\frac{\delta}{3}$. Therefore, the average linkage will combine all nodes belonging to the same role before it links nodes that belong to different roles. □

**Corollary E.1.** *Let $A^{(1)}, ..., A^{(s)}$ be independent samples of the RIP model with the same role assignment ($\Omega_{role}$ must not necessarily be the same). Assuming the prerequisites of theorem 4.3 for $A = \frac{1}{s} \sum_{i=1}^{s} A^{(i)}$ - except eq. 8. If*

$$s > -\frac{9\mathcal{W}_{-1}((q-1)\delta^2/9k^2)}{2n\delta^2}$$

*Then with probability at least q: Algorithm 1 finds the correct role assignment using average linkage for clustering.*

