# OpenReview forum: "An Optimization-based Approach To Node Role Discovery in Networks: Approximating Equitable Partitions"
_NeurIPS.cc/2023/Conference — NeurIPS 2023 poster_

### Official Review · Reviewer_rRxj · 2023-07-06

**Soundness:** 3 good
**Presentation:** 2 fair
**Contribution:** 3 good
**Rating:** 5
**Confidence:** 3

**Summary:**

The paper offers a new perspective on the problem of role extraction while defining node roles based on the ideas of equitable partitions and and graph-isomorphism tests, the Weisfeiler-Leman algorithm. The paper studies two associated optimization problems (cost functions) inspired by graph isomorphism testing and provides theoretical guarantees for their solutions.  To validate the approach, a novel "role-infused partition benchmark" is introduced, enabling the generation of networks where nodes possess different roles in a stochastic manner.

**Strengths:**

(+) The paper offers a new perspective on node role extraction, focusing on equitable partitions and providing a principled stochastic notion of node roles. This approach adds to the existing methods in the field and offers a new way to define and quantify node roles in complex networks.

(+) The paper presents a quantitative definition of node roles based on equitable partitions, allowing for the numerical measurement of deviation from an exact equivalence. This provides a more nuanced understanding of node roles compared to traditional definitions.

(+) The technical aspects of the paper are thorough and detailed. Also, the technical details aeem to be correct.

(+) Numerical experiments show the effectiveness and superiority of the proposed method over a graph neural network, GIN, for three different tasks.

**Weaknesses:**

(-) While the paper briefly mentions different categories of node role extraction approaches (graphlet-based, walk-based, and matrix-factorization-based), it does not provide a detailed comparison or analysis of how the proposed equitable partitions approach compares to these existing methods. A more rigorous comparative analysis, including performance metrics and evaluation on benchmark datasets, would strengthen the paper's contribution and demonstrate the advantages of the proposed approach.

(-)  The idea of Equitable partitions for node role discovery is interesting. However, the details regarding "why this approach" makes sense is missing. There can be other ways too? Why this method should work? At a conceptual level, a more rigorous explanation related to the definition of roles based on equitable partition is missing. I think this is crucially important. In other words, the paper focuses on providing details of "designing" and approach based on EP, while leaving details of "why this is appropriate"

(-) A more thorough literature review might be needed. For instance, the following paper provides a nice simple algorithm for computing equitable partitions. (I am not sure it is better or worse; however it might be handy to have a look at it.)

- Zhang et al., "Upper and Lower Bounds for Controllable Subspaces of Networks of Diffusively Coupled Agents," IEEE Transactions on Automatic control, 2014.

Also, there is some recent work on counting subgraphs (that may define node roles based on the structural attributes of the graph), for instance,

- Hassan et al., "Computing Graph Descriptors on Edge Streams." ACM Transactions on Knowledge Discovery from Data, 2023.

(-) The paper introduces a family of cost functions to assess the quality of a role partitioning. However, it does not thoroughly discuss the selection, design, or properties of these cost functions. Providing a more in-depth exploration and analysis of the different cost functions, their properties, and how they relate to the problem of node role extraction would enhance the technical understanding of the proposed approach.

(-) It is unclear how the ground truth for the sampled adjacency matrices is computed  in Experiment 1 of Section 5. Moreover, GIN is a relatively old graph neural network. There are recent methods that show better results on several downstream tasks and could have been considered for baseline comparisons. Some of these works include the work of Bouritsas et al, and Ramp\'a\v{s}ek et al, below.

- Bouritsas et al., "Improving graph neural network expressivity via subgraph isomorphism counting," IEEE Transactions on Pattern Analysis and Machine Intelligence. 2022 Feb 24;45(1):657-68.

- Ramp\'a\v{s}ek, et al., "Recipe for a general, powerful, scalable graph transformer," Advances in Neural Information Processing Systems. 2022.

**Questions:**

(-) It is mentioned that Equitable partitions imply an upper bound on the expressivity of Graph Neural Networks while citing references [26] and [43]. There is nothing of that sort in these papers. Could you point out where such results are mentioned in the said references.

(-) As coarsest equitable partition is an important concept for the paper, an example to illustrate this concept or a proper reference to another paper is vital and missing in the draft.

(-)  As mentioned in the "weakness", can you explain more rigorously the motivation for using equitable partitions for node role discovery.

(-)  In Theorem 3, what is the interpretation of the parameter $\delta$ and what is the computational complexity to calculate it?

(A minor comment) There are minor grammatical errors, e.g.,

-  "Thus, from any node $v$ within in the same class ..." (in is not needed).

- "This also allows for Gradient Descent approaches like e.g. GNNs." (like and e.g., are used simultaneously).



**Limitations:**

The limitations and possibly approaches to tackle these limitations are presented in the paper.

---

> ### Author Rebuttal · Authors · 2023-08-09
>
> Dear Reviewer rRxj,
>
> Thank your for your review.
>
> Regarding your questions:
> 1. Expressivity of GNNs: The cited references do, in fact, prove the statement. The contraposition of Lemma 2 in (Xu et. al [43]) yields "If the WL test decides G_1 and G_2 are isomorphic, then any GNN maps G_1 and G_2 to the same embedding.". (Morris et al. [26]) Thm 1 is more specific stating that the color $c^{(t)}(v)$ assigned to node $v$ at iteration $t$ of the WL algorithm encodes "more" information than the features $f^{(t)}(v)$ of a node in that: "$c^{(t)}(v) = c^{(t)}(u) \implies f^{(t)}(v) = f^{(t)}(u)$". By the refinement property of WL, we have that the last iteration $T$ holds the "most information" and hence $c^{(T)}(v) = c^{(T)}(u) \implies f^{(t)}(v) = f^{(t)}(u) \forall t \leq T$.
>
> 2. References to ceP: We provide references to multiple papers that use equitable partitions in multiple ways: Graph isomorphism testing [3,4,25,41] reduction of complex systems [24,30,36,37,44] and Graph Neural Networks [26,43]. The notion of equitable partitions emerged somewhat independently in many different parts of research such as social sciences in the study of social networks, electrical engineering as a means of model order reduction, and theoretical computer science in the form of the WL test. The idea is the same, but the formalisms vary quite substantially from the color perspective of the WL test to the matrix equation adopted by the model order reduction community (and us in this paper).
>
> 3. Motivation for using EPs: We motivate the use of the cEP for role extraction in section 2.1 as a means of model order reduction using the quotient graph and in section 4 by highlighting that even graph neural networks cannot extract more structural information from a graph than the cEP. We would of course like to go even further into the motivation, and will include it if space permits: The WL colors (and by extension the cEP) carry an immense amount of information about the structure of the network surrounding a node. One can think of it as iteratively building a tree of possible paths from the start node to neighboring nodes. At the first iteration, the color $c^{(t)}(v)$ encodes the degree of $v$ in the graph. At the second iteration, it encodes the number of nodes with a certain degree that are adjacent to $v$ and so on. This tree of possible paths is profoundly important for processes that move over the edges of a network such as dynamical systems and GNNs, and even simple ones like PageRank and other centrality measures. However, the colors are often too descriptive which has recently sparked a series of investigations on how to meaningfully simplify this immense descriptive power [6,16,19] (and this paper).
>
> 4. Parameter $\delta$: The parameter $\delta$ in Theorem 3 is a parameter that determines how hard it is to recover a given SBM. This bares some resemblance to standard SBM inference techniques, that consider a threshold for which exact recovery is possible (Abbe, Emmanuel, and Colin Sandon. "Community detection in general stochastic block models: Fundamental limits and efficient algorithms for recovery." 2015 IEEE 56th Annual Symposium on Foundations of Computer Science. IEEE, 2015). Intuitively, the roles (or blocks in the original SBM) may not be too similar, otherwise it takes many samples to see a significant difference. $\delta$ describes the minimal difference of the roles in the ground truth model. $\delta$ is easy to compute given the ground truth model. One simply computes the WL colors ($O(n^2*log(n))$) and computes the minimal difference in intermediate roles ($O(n^2)$).
>
> 5. Thanks for pointing out the typos! At some point one stops seeing them.
>
> Weaknesses:
> 1. Literature review: The notion of **almost** equitable partitions partitions in (Zhang et al., "Upper and Lower Bounds for Controllable Subspaces of Networks of Diffusively Coupled Agents," IEEE Transactions on Automatic control, 2014.) is sometimes also called **externally** equitable partition, because it only takes into account edges between distinct classes of the partition. On those, however, it is an *exactly* externally equitable partition. It is often useful as it gives an upperbound on the controllability of a system similar to the upper bound that is given by exactly equitable partitions for GNNs. However, this does not coincide with the notion presented in this paper: Contrary to the latter notion, our **approximately** equitable partitions do not have to be exact, but may only consist of $k$ classes.
>
> 2. GIN as baseline. While the GIN is relatively old, it is still often used as a baseline - and is provably among the most expressive message passing GNNs there are [43]. Due to this fact, it is more than sufficient for our purpose here. GNNs like the one you proposed would also "break" the WL upper bound, which improves expressiveness but isn't desirable in this scenario - as discussed in the appendix.
>
> Thanks again for your review.
>
> Best,
> the authors

---

### Official Review · Reviewer_nMtV · 2023-07-06

**Soundness:** 2 fair
**Presentation:** 3 good
**Contribution:** 3 good
**Rating:** 5
**Confidence:** 3

**Summary:**

This paper presents a novel perspective on the problem of node role extraction in complex networks, highlighting its distinctions from community detection. The authors propose a definition of node roles and introduce two optimization problems based on graph-isomorphism tests, the Weisfeiler-Leman algorithm, and equitable partitions. Theoretical guarantees are provided, and the approach is validated using a newly introduced "role-infused partition benchmark" that allows for stochastic assignment of different roles to nodes in sampled networks. The findings contribute to network analysis, graph mining, and the development of reduced order models for dynamical processes on networks.

**Strengths:**

- The research topic, role discovery in graphs, is very important and is worth studying, especially compared to the community detection problem, role discovery is also important but lacks attention.

- The design of the model based on equitable partitions is technically sound and theoretically guaranteed.

- Experimental results on several datasets from different perspectives show the effectiveness of the proposed method.

**Weaknesses:**

- Baseline selection. Some relevant and/or important baselines have not been compared.

- Downstream tasks. The proposed method is not tested on widely used downstream tasks to show its effectiveness.

- Dataset. There are some more widely used datasets for role discovery that are not used in this paper.

**Questions:**

Most of my questions are related to the experiments:
- There are some relevant baselines that are not compared, for example, MMSB [1] and RIDRs [2]. The first one you can also discuss the relation with your proposed method, e.g., if it is possible to extend the proposed method to soft role assignment like MMSB. The second method is also based on equitable partitions.

- There are some commonly used datasets that can be used for evaluating role discovery, for example, these datasets used in [3].

- In previous role discovery studies, different downstream tasks have been selected to show the effectiveness of the discovered roles, e.g., node classification and link prediction. For example, these tasks in RolX [4] and struc2vec [3]. Also, there is no evaluation of the discovered roles. For example, in RolX, the quantitative evaluation using clustering metric has been used.

- Efficiency issue. Although the proposed method is technically sound, a question is how efficient the proposed method is, e.g., how large a graph this method can handle.

[1] Airoldi E M, Blei D, Fienberg S, et al. Mixed membership stochastic blockmodels[J]. Advances in neural information processing systems, 2008, 21.

[2] Gupte P V, Ravindran B, Parthasarathy S. Role discovery in graphs using global features: Algorithms, applications and a novel evaluation strategy[C]//2017 IEEE 33rd International Conference on Data Engineering (ICDE). IEEE, 2017: 771-782.

[3] Henderson K, Gallagher B, Eliassi-Rad T, et al. Rolx: structural role extraction & mining in large graphs[C]//Proceedings of the 18th ACM SIGKDD international conference on Knowledge discovery and data mining. 2012: 1231-1239.

[4] Ribeiro L F R, Saverese P H P, Figueiredo D R. struc2vec: Learning node representations from structural identity[C]//Proceedings of the 23rd ACM SIGKDD international conference on knowledge discovery and data mining. 2017: 385-394.

==============================================

After the rebuttal, I increased my overall rating.


**Limitations:**

No potential negative societal impact.

---

> ### Author Rebuttal · Authors · 2023-08-09
>
> Dear Reviewer WiNy,
>
> Thank your for your review.
>
> Regarding your questions:
> 1. MMSBM and RID$\epsilon$R. While the comparison with RIDR does seem interesting, we were unable to find an implementation for neither RIDR nor inference of the mixed membership SBM. If time permits, we would like to also include RIDR in our comparison. Regarding the MMSBM, the soft assignments do not change the fact, that the MMSBM finds communities. Suppose in the RIP model, $p=0, c = k > 2, \Omega_{role} > 0$. Then, nodes only have edges inside the clusters and all "outside edges" are not there. For $n \rightarrow \infty$, the likelihood of placing two nodes from the same ground-truth community in distinct blocks of the MMSBM (even only fractionally) goes toward $0$, thus the MMSBM will recover the communities and not the roles.
>
> 2. We have conducted new experiments on the proposed datasets, please refer to the general rebuttal and attached pdf.
>
> 3. We would have liked to also encompass the RolX datasets in our experiment, sadly only the raw data is linked in the paper, which they modified (added the targets) for their experiments. However, the performance should be similar to that presented in struc2vec.
>
> 4. Our proposed methods are exceedingly efficient. The optimization algorithm for the long-term cost function (EV algorithm) uses $O(nk + n\log(n))$ time. The time needed by the approximate WL algorithm is largely dominated by the clustering used. Other than that, the algorithm only consists of $k$ matrix-vector products resulting in a complexity of $O(i*(E*k+C))$, where $i$ is the number of iterations, $E$ the number of edges, and $C$ the time for clustering.
> We have also conducted an experiment of real world time consumption in the attached pdf of the general rebuttal.
>
> Once again, thank you for your review.
>
> Best,
> the authors

---

> > ### Comment · Reviewer_nMtV · 2023-08-15
> > **Thanks for the responses**
> >
> > Dear authors,
> >
> > Thanks for your efforts on the rebuttal. These answers addressed my concerns. I thereby increase my ratings.

---

> > > ### Author Response · Authors · 2023-08-17
> > > **Thanks for your response**
> > >
> > > Dear Reviewer nMtV,
> > >
> > > Thanks for your response!
> > >
> > > Cheers,
> > > the authors

---

### Official Review · Reviewer_WiNY · 2023-07-06

**Soundness:** 3 good
**Presentation:** 3 good
**Contribution:** 2 fair
**Rating:** 6
**Confidence:** 4

**Summary:**

The paper considers a relaxed definition of the coarsest equitable partition (CEP), which equals the final partition of Weisfeiler-Leman or the color refinement algorithm in the original version. The authors allow to specify the number of cells of the partition and derive a related optimization problem. From this, an algorithm for role discovery is derived, outperforming techniques such as role2vec on real-world graphs.

**Strengths:**

* The concept of structural roles is closely related to the Weisfeiler-Leman algorithm and fundamental for graph learning.
* The relaxation of the final WL partition with a fixed number of cells is novel.
* The RIP model for graph generation is innovative.

**Weaknesses:**

* Algorithm 1 is not analyzed sufficiently. It would be interesting to have approximation guarantees. Moreover, it would be nice to investigate the case where $k$ equals the number of cells in the cEP. I believe that the current version would not guarantee that the cEP is found since the cluster method obtains only $X$ as an argument neglecting $H$, which should be required to guarantee this. Is this correct?
* Several other methods compute relaxed WL partitions without allowing to specify $k$ and are discussed in section 2.1. These could also be used in the experimental comparison to assess the advantages of the proposed method.

**Questions:**

* Could you explain whether Algorithm 1 can recover the cEP for a suitable choice of the parameter $k$?

**Limitations:**

Limitations are sufficiently discussed.

---

> ### Author Rebuttal · Authors · 2023-08-09
>
> Dear Reviewer WiNy,
>
> Thank your for your review.
>
> Regarding your question, the results of Algorithm 1 heavily depend on the clustering algorithm used. If one was to use a clustering algorithm that does not assign two data points with the same value to different clusters, then Algorithm 1 would output the cEP.
> For example, assume the clustering algorithm used returns the same cluster for data points with the same value and different clusters for data points with different values (it may return as many clusters as it wants). Then one recovers the adapted Weisfeiler-Lehman algorithm that Morris et al. [1] used in the proof of Theorem 2. This algorithm uses only the multiset of neighboring colors (and not the previous color of the node) but is equivalent to the "full" WL algorithm. In the same way it is sufficient to only cluster X without access to H and still arrive at the cEP.
>
> Indeed, you can think of it this way (although this was out of scope for this particular paper): The columns of the indicator matrix H_EP of the cEP are a basis of a subspace that is invariant to multiplication with A and the starting vector (all-ones) is an element of this subspace. Thus step 4 can never leave this subspace and if we assume that the clustering step (5) also does not break this, then we never leave the subspace. Once we cannot divide the clusters any further, we have reached the cEP.
>
> Weaknesses:
> Guarantees for Algorithm 1 are extremely dependent on the clustering algorithm used. Most clustering algorithms offer no guarantees on performance which makes it exceedingly hard to provide guarantees for Algorithm 1.
> We even have hardness of approximation - as is the case for the closely related kmeans. Indeed, combining our reduction from k-means with the hardness of approximation of Pranjal et al. [2] results in a similar "hardness of approximation" statement for the short-term cost. However, in our reduction the dimension is encoded through additional 'predetermined' clusters and their reduction results in a very high dimension of the vectors. Thus, this holds only for large values of $k$ whereas we are mostly interested in small values of $k$.
>
> The other approaches specified in section 2.1 all do not permit to set a specific number of clusters - the main difference of our approach to  [3] is precisely this. They specify an $\epsilon$ deviation that the final clustering must uphold and then aim to find the coarsest partition (which is NP-hard also). However, the number of classes is only indirectly determined by this. It may be that there exists no $\epsilon'$ such that the number of clusters found is exactly $k$. So even searching along the values of $\epsilon'$ would not mean that we can get exactly $k$ roles. Regarding our cost function, it will always be beneficial to have more rather than fewer clusters. This is why comparing the two approaches is extremely difficult.
>
>
> Thanks again for your review!
>
> Best,
> the authors
>
> [1] Morris, Christopher, et al. "Weisfeiler and Leman Go Neural: Higher-order Graph Neural Networks." arXiv preprint arXiv:1810.02244 (2018).
>
> [2] (Awasthi, Pranjal, et al. "The hardness of approximation of euclidean k-means." arXiv preprint arXiv:1502.03316 (2015).)
>
> [3] (M. Kayali and D. Suciu. Quasi-stable coloring for graph compression: Approximating max-flow, linear programs, and centrality. arXiv preprint arXiv:2211.11912, 2022.)

---

> > ### Comment · Reviewer_WiNY · 2023-08-16
> >
> > Thanks for answering my question. The link to the adapted WL in [1] was very helpful. Please consider addressing this question also in the paper. I have increased my score accordingly.

---

> > > ### Author Response · Authors · 2023-08-17
> > > **Thanks for your response**
> > >
> > > Dear reviewer WiNY,
> > >
> > > Thanks for your response! If space permits, we will add this perspective on our algorithm to the final version of our paper.
> > >
> > > Best,
> > > the authors

---

### Official Review · Reviewer_ZAin · 2023-07-07

**Soundness:** 3 good
**Presentation:** 4 excellent
**Contribution:** 3 good
**Rating:** 6
**Confidence:** 3

**Summary:**

The authors propose the notion of equitable partitions from graph isomorphism literature in order to partition the nodes of a network according to their structural roles. They study two optimization problems for approximately recovering such equitable partitions. Analogous to the SBM model, the authors devise the RIP (role-infused partition) stochastic model and validate their findings on this model.

**Strengths:**

The paper is very well-written. The proposed cost functions for computing approximate equitable partitions are very well-motivated and natural. Approximate versions of the Weisfeiler-Leman algorithm is an important question which deserves much study. The proposed RIP model is a very satisfcatory benchmark for role discovery.

**Weaknesses:**

The experimental work is inadequate in order to understand the impact of the proposed formalism to real-world data (Section 5, Experiment 3). The considered datasets are too few, specialized and small-sized in order to deduce that approximate equitable partitions (and approximate Weisfeiler-Leman) is at the core of role discovery.

**Questions:**

1) On the considered molecular datasets, does the approximate number of equitable partitions differ (the parameter "k" \in {2,20}) differ significantly from the actual number of equitable partitions, on an average? Since the average number of nodes is 30, and since I would expect most graphs to have discrete equitable partitions (i.e. one node per class), it will be interesting to know if the optimal values of k were indeed small (say 2-6).

**Limitations:**

Yes.

---

> ### Author Rebuttal · Authors · 2023-08-09
>
> Dear Reviewer ZAin,
>
> Thank you for your review.
>
> Regarding your question, one has to keep in mind that we are concerned with finding "global" roles. That is, we consider the whole dataset as one graph with multiple connected components. On the proteins dataset, there are $1278$ classes of the *exact* cEP. This would be the optimal value for k since it would result in a cost of $0$ - and every $k<1278$ has an optimal cost $c>0$.
> If we restrict ourselves to $k \in \{ 2,20 \}$ (as we did for the experiments), then we cannot make a statement about the optimal value of the cost function. The problem does not decompose into finding the best approximate EP for each connected component individually. And as the problem is NP-hard we have no hope in brute forcing it for this size of the graph.
>
> Regarding our experiments, please also take a look at the general author response.
>
> Thank you once more for your review.
>
> Best,
> the authors

---

> > ### Comment · Reviewer_ZAin · 2023-08-19
> >
> > Thanks for your response. After going through the rebuttal and other responses, I have decided to maintain my original score.

---

### Author Rebuttal · Authors · 2023-08-09

Dear Reviewers,

Thank you all for your reviews.

A common theme in your reviews was a need to broaden the experimental part of the paper. We have thus extended our experiments by two experiments proposed by Reviewer nMtV and originally conducted by [1]. In particular, this includes the downstream task of classifying airports due to their throughput of passengers into 4 categories - the 4 roles.

Before going into detail we would like to note a few things: First, it is extremely hard to find data on which roles can be evaluated down-stream. In most role extraction approaches (struc2vec, rolX), the target classifications are constructed either from metadata or from some characteristic of the graph which is then assumed to correspond to the role. However, it is not easy to check if this assumption is correct. This is one of our main contributions: Giving a rigorous definition of "role" and a benchmark generative model on which to assess these roles. Our other main contribution is providing multiple inference algorithms and ways to assess the "goodness" of such a role assignment.

While it was not our main goal in this paper to provide an embedding that can be used on some downstream task, we performed two new experiments, based on your feedback. Toward this, we used an embedding based on our role assignment:
For a given adjacency matrix $A$ with dominant eigenvector $\nu$, we compute the output of the approximate WL algorithm $H$ using average linkage. We also compute the output $H_{EV}$ of the EV algorithm. As the embedding of each node, we use:
$$emb(i) = (H[i,:], (H_{EV})[i,:], (AH)[i,:], (AH_{EV})[i,:], \nu[i])$$
In other words, we embed the nodes with the short-term and long-term role assignment, the adjacencies with other roles and the component of the eigenvector corresponding to that node. For this dataset, we used $k=30$ roles on all datasets. We set $k$ by hand due to time, but want to automatically tune it for the final paper to improve performance. In all other regards, we copied the experimental design of [1], that is, we fit a "one vs all" logistic regression on the embeddings and perform a 5-fold crossvalidation tuning hyperparameters on every fold independently.
The results can be seen in the attached pdf.

We look forward to the discussion!

Best,
the authors

[1] Ribeiro, Leonardo FR, Pedro HP Saverese, and Daniel R. Figueiredo. "struc2vec: Learning node representations from structural identity." Proceedings of the 23rd ACM SIGKDD international conference on knowledge discovery and data mining. 2017.

---

### Decision · Program_Chairs · 2023-09-21

**Decision:**

Accept (poster)

**Comment:**

The reviewers raised a few concerns, including the experimental work and algorithm analysis, etc., on this paper. The authors addressed some concerns during the rebuttal/discussion phase, and finally the reviewers were relatively positive about the work.
After reading the reviews and discussions, and also taking the scores given into account, I am recommending this paper be accepted as a poster.